

# Co-expression network analysis reveals the pivotal role of mitochondrial dysfunction and interferon signature in juvenile dermatomyositis

Danli Zhong, Chanyuan Wu, Jingjing Bai, Dong Xu, Xiaofeng Zeng and Qian Wang

Department of Rheumatology, Peking Union Medical College Hospital, Peking Union Medical College & Chinese Academy of Medical Sciences; National Clinical Research Center for Dermatologic and Immunologic Diseases (NCRC-DID), Beijing, China
Key Laboratory of Rheumatology and Clinical Immunology, Ministry of Education, Beijing, China

Corresponding author
Qian Wang, wangqian@pumch.cn

## ABSTRACT

**Background**. Juvenile dermatomyositis (JDM) is an immune-mediated disease characterized by chronic organ inflammation. The pathogenic mechanisms remain ill-defined.
**Methods**. Raw microarray data of JDM were obtained from the gene expression omnibus (GEO) database. Based on the GSE3307 dataset with 39 samples, weighted correlation network analysis (WGCNA) was performed to identify key modules associated with pathological state. Functional enrichment analyses were conducted to identify potential mechanisms. Based on the criteria of high connectivity and module membership, candidate hub genes were selected. A protein-protein interaction network was constructed to identify hub genes. Another dataset (GSE11971) was used for the validation of real hub genes. Finally, the real hub genes were used to screen out small-molecule compounds via the Connectivity map database.
**Results**. Three modules were considered as key modules for the pathological state of JDM. Functional enrichment analysis indicated that responses to interferon and metabolism were dysregulated. A total of 45 candidate hub genes were selected according to the pre-established criteria, and 20 genes could differentiate JDM from normal controls by validation of another external dataset (GSE11971). These real hub genes suggested the pivotal role of mitochondrial dysfunction and interferon signature in JDM. Furthermore, drug repositioning highlighted the importance of acacetin, helveticoside, lanatoside C, deferoxamine, LY-294002, tanespimycin and L01AD from downregulated genes with the potential to perturb the development of JDM, while betonicine, felodipine, valproic acid, trichostatin A and sirolimus from upregulated genes provided potentially therapeutic goals for JDM.
**Conclusions**. There are 20 real hub genes associated with the pathological state of JDM, suggesting the pivotal role of mitochondrial dysfunction and interferon signature in JDM. This analysis predicted several kinds of small-molecule compounds to treat JDM.

## INTRODUCTION

Juvenile dermatomyositis (JDM) is a rare chronic childhood-onset autoimmune disease characterized by inflammatory infiltration in small vessels and tissues within skin and muscle. The incidence of JDM is 2–4 per million per year in the United States (*Feldman et al., 2008*), with female:male ratios ranging from 1.5:1 to 5:1 (*Lindsley et al., 1995*). The major manifestations of JDM patients consist of symmetrical proximal muscle weakness, skin rashes, and internal organs involvement (*Crowe et al., 1982*). Up to 30% of JDM may present with calcifications, one of the prognostic factors of long-term disability (*Arabshahi et al., 2012*; *Li & Zhou, 2019*; *Ravelli et al., 2010*). Adults with JDM in childhood are susceptible to premature cardiovascular damage (*Gitiaux et al., 2016*).

Pathological state and treatment have been reported to affect growth and puberty in the active phase of JDM (*Nordal et al., 2019*). Ongoing disease activity, irreversible damage, and aggressive immunosuppressive therapy remain major challenges for long-term outcomes and quality of life in JDM patients (*Hoeltzel et al., 2014*). The etiology of JDM remains ill-defined although genetic and environmental factors are suspected to be involved in its pathogenesis. It has been reported that JDM patients had higher incidence of Epstein-Barr virus infection (*Zheng et al., 2019*), and the prominent type 1 interferon (IFN) signature was shown to affect the vasculature JDM (*De Paepe, 2017*; *Greenberg, 2010*). Adaptive and innate immune mechanisms involving IFN-associated molecules appear to mediate endothelial tubule-reticular formations and peri-fascicular atrophy.

Weighted gene co-expression network analysis (WGCNA) algorithm is a powerful bioinformatic method that mines practical information from gene expression profiles by constructing of gene modules, thereby interpreting the biological significance of a gene (*Langfelder & Horvath, 2008*). WGCNA has been widely used in various diseases (*Zhao et al., 2010*), including malignancies, cardiovascular diseases and autoimmune diseases, where it has provided useful information for understanding pathological process and for discovery of diagnostic and prognostic biomarkers. Nevertheless, WGCNA has never been applied to JDM.

Therefore, we used WGCNA for the first time to analyze pathological state and gene expression data in JDM muscular samples to explore and validate hub genes associated with JDM, as well as to predict small-molecule compounds to treat JDM with promising perspectives.

## MATERIALS & METHODS

### Data collection and differentially expressed genes screening

The flowchart of the study is shown in Fig. S1. Microarray profiles of JDM were retrieved from the Gene Expression Omnibus (GEO, http://www.ncbi.nlm.nih.gov/geo/) of the National Center for Biotechnology Information using the search terms of "juvenile dermatomyositis" restricted in the title. The datasets enrolled in this study must contain musclular specimens with three biological replicates at least. The "affy" package in R environment (version 3.6.1) was used to quantile normalize the expression within each dataset (*Sasik, Calvo & Corbeil, 2002*). The corresponding platforms were applied

to annotate each probe according to Entrez ID, and the average expression value was calculated if several probes corresponded to the same Entrez ID (Table S1). The "limma" R package was performed for identifying differentially expressed genes (DEGs) between JDM samples and normal samples under cut-off criteria of false discovery rate (FDR) <0.05 and |log2fold change| ≥ 1.

## Co-expression network construction

The variance of each gene expression value was calculated and the genes with variance ranked in the top 25% were selected for the construction of WGCNA (*Langfelder & Horvath, 2008*). The "WGNCA" package was used to construct the co-expression network. In detail, the function goodSamplesgenes was used to include the qualified genes and samples, followed by choosing an appropriate soft-thresholding power to construct the weighted adjacency matrix by the function pickSoftThreshold. The adjacency matrix was transformed into the topological matrix (TOM), and TOM-based dissimilarity (1-TOM) measure was used to cluster the genes using the flashClust function. Genes in the same module were highly interconnected. Then, phenotype (clinic traits) was imput into the co-expression network, and the following parameters were calculated: module eigengene (ME), gene significance (GS), and module membership (MM). ME represents the significant component in the principal component analysis for each gene module, and MM refers to the connectivity between genes and modules. GS was representative of correlation strength between gene expression and clinical traits, which was calculated by log10 transformation of the *P*-value (GS = lg P) in the linear regression. Key modules were considered based on the criteria that the correlation coefficient ≥ 0.80 and *P*-value <0.05.

## Functional enrichment analysis

All genes in key modules were uploaded to the g:Profiler online (*Reimand et al., 2007*) database to perform Gene Ontology (GO) functional annotation (*Ashburner et al., 2000*) and the Kyoto encyclopedia of genes and genomes (KEGG) enrichment pathway analysis (*Kanehisa & Goto, 2000*). GO functional analysis consists of biological process (BP), cellular component (CC), and molecular function (MF). Analysis results were extracted under the condition of adjusted *P*- value <0.05. The top five terms were visualized if there were more than five terms.

## Selection and validation of hub genes

Genes with high correlation in candidate modules were defined as candidate hub genes. High connectivity was considered when the connectivity ranked in the top 2%. Candidate hub gene met the absolute values of MM >0.80 and GS >0.20. After identifying hub genes highly associated with clinical traits, the search tool for the retrieval of interacting genes (STRING) database was used to construct a protein-protein interaction (PPI) network for the candidate hub genes, and molecular complex detection (MCODE, a plugin in Cytoscape) was used to further select the real hub (*Shannon et al., 2003*; *Szklarczyk et al., 2015*). Genes with MCODE score ≥ 0 in the PPI network were selected as the final hub genes. A separate dataset (GSE11971) was used to validate the differential expression of the final hub genes.

### Related small-molecule compounds screening

Connectivity map (CMap) database (https://portals.broadinstitute.org/cmap) was used to screen out small molecule compounds based on the real hub genes associated with JDM, because most compounds in this database are the United States Food and Drug Administration-approved drugs (*Lamb et al., 2006*). First, real hub genes were divided into upregulated and downregulated groups. Next, these probe sets were used to query the CMap database based on the platform of the Affymetrix Human Genome U133 Plus 2.0 Array (http://www.affymetrix.com/analysis/netaffx/index.affx). Finally, enrichment scores representing similarity were calculated, ranging from −1 to 1. Small molecules generated from up-regulated genes suggested therapeutic goals, while down-regulated genes predicted inhibitors of therapy for the disease. Potential compounds were selected based on connectivity score, *P*-value and correlation.

### Statistical analysis

Two-tailed Student's *t*-test was applied to the significance of differences between groups, and *P*-value less than 0.05 was considered as statistically significant. Statistical analyses were performed using Graphpad Prism 8.0.

## RESULTS

### Data collection and differentially expressed genes

We employed two datasets on JDM muscular expression profiles. Dataset GSE3307 was used as the training set (*Bakay et al., 2006*). The original study enrolled 39 muscular biopsy samples, including 21 JDM patients and 18 healthy controls (HC). Dataset GSE11971, including nineteen JDM patients and four normal controls, was used as the validating set (*Chen et al., 2008*). The gene expression profiles of all tissue samples were analyzed based on the platform of the Affymetrix Human Genome U133 Plus 2.0 Array. A total of 2,834 differentially expressed genes between JDM and HC were identified, including 1,888 down-regulated genes and 946 up-regulated genes. The DEGs are listed in Table S2.

### Construction of a weighted co-expression network and identification of key modules

5103 genes whose variance ranked in the top 25% with 21 JDM samples and 18 control samples in GSE3307 were used for WGCNA construction. The "WGCNA" R package was used for expression matrix of GSE3307, and soft-thresholding power $\beta$ value equal to 10 was selected to ensure a scale-free network with scale-free $R^2$ equal to 0.90 (Figs. S2A–S2B) (*Langfelder & Horvath, 2008*). A total of 13 modules were returned by WGCNA analysis (Figs. 1A–1B).

The interaction relationship of 12 modules was analyzed using network heatmap plots (Fig. 1C). The division of all modules was highly independent from our analysis. The module eigengene dendrogram showed that 12 modules were divided into two clusters, and the adjacency heatmap of eigengene showed a similar result (Fig. 2A). Based on the criteria that correlation coefficient $\geq$ 0.80, *P* value <0.05, blue, lightgreen and midnightblue modules were identified as key modules for further analysis (Fig. 2B). Therefore, we

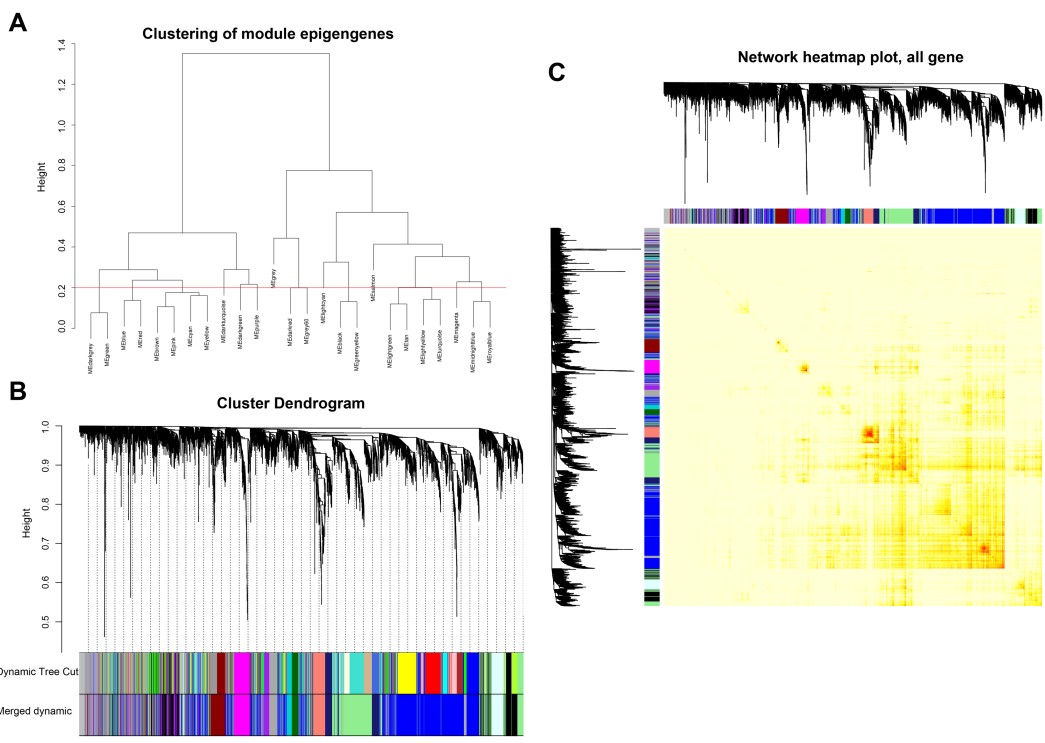

**Figure 1** **Construction of WGCNA modules.** (A) Dendrogram of module eigengenes based on dissimilarity measure (1-TOM); (B) Cluster dendrogram of the genes with variance ranked in the top 25% in the GSE3307 dataset. Each color represents one module; (C) Network heatmap plots of genes selected for WGCNA construction. The depth of yellow in the middle of the figure indicated the degree of correlation between pair-wise genes.

selected the blue, lightgreen and midnightblue modules for subsequent analysis, to identify the relevance between key modules and the pathological state of JDM with substantial biological significance (Figs. 2C–2E).

## Functional and pathway enrichment analysis

GO and KEGG pathway enrichment was performed for all genes in the key modules to mine the biological functions associated with JDM. Biological process of GO analysis showed genes in the blue module were associated with generation of SRP-dependent cotranslational protein targeting to membrane, cotranslational protein targeting to membrane, protein targeting to ER, nuclear-transcribed mRNA catabolic process and establishment of protein localization to endoplasmic reticulum; and that in the lightgreen module was relevant to response to type I interferon, type I interferon signaling pathway, cellular response to type I interferon, defense response to virus and response to virus.The top five pathways related to the midnightblue module were cellular response to chemical stimulus, extracellular structure organization, extracellular matrix organization, response to organic substance and cell motility (Fig. 3A). Pathway enrichment results of MF and CC in three key modules are presented in Figs. 3B–3C. The results of the KEGG pathway enrichment analysis in three modules are shown in Fig. 3D.

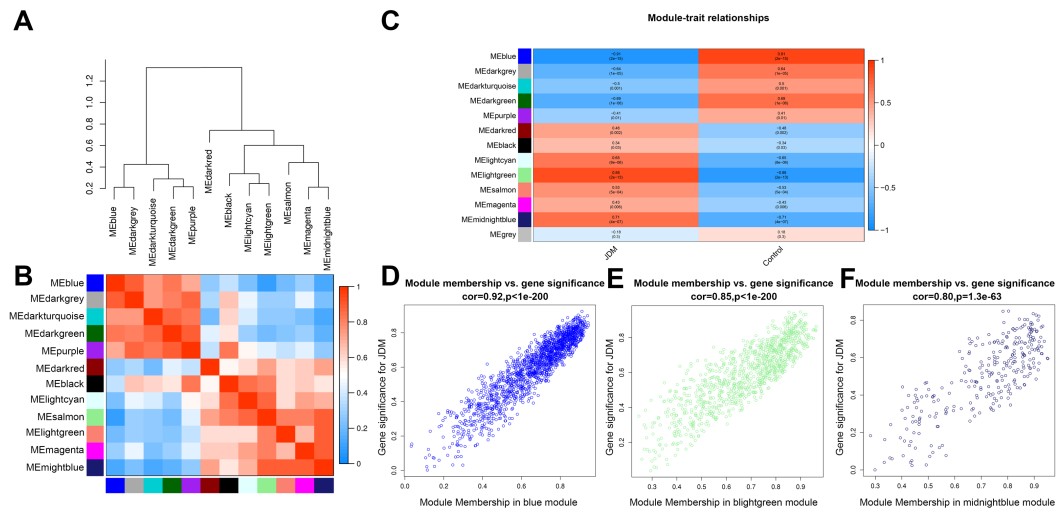

**Figure 2** **Relationship between modules and clinical traits.** (A) Module eigengene dendrogram and (B) Adjacency heatmap. (C) Heatmap of the module-trait relationships. (D–F) Scatter plots of module eigengenes in blue (C), lightgreen (D) and midnight blue (E) modules.

## Identification of hub genes

Based on the criteria that MM >0.80 and GS >0.20, a total of 45 DEGs with the high connectivity in key modules were screened as candidate hub genes. Then, a PPI network was constructed for candidate hub genes using Cytoscape, consisting of 42 nodes and 80 edges according to STRING database (Fig. 4). We conducted molecular complex detection (MCODE) (a plugin in Cytoscape) analysis for 45 candidate hub genes, and 28 genes (blue = 15, lightgreen = 11, midnightblue = 1) were considered hub genes according to the criteria of MCODE score ≥ 0. Table 1 shows 28 hub genes in the three modules.

All hub genes were validated using JDM data from another GEO database (GSE11971). Because of the differences in microarray probes used in two data sets, boxplots were used to show the validation results for the final 22 hub genes (Fig. S3). We found that seven genes, SP110, SAMHD1, IFIT5, PLSCR1, IFI16, MX2 and CLIC1, were significantly upregulated in JDM compared to HC, while thirteen genes, COX5B, COX6A2, COX7C, NDUFA4, NDUFB4, MDH2, ATP5O, ATP5B, RPL21, TPI1, SLC25A3, VDAC1 and EIF4B, were significantly downregulated in JDM in comparison of HC. Figure 5 summarizes the cross-talk pathways involved in the pathogenesis of JDM by hub genes and literature (*Miller et al., 2018*; *Thompson, Piguet & Choy, 2017*).

## Related small-molecule compounds screening

The CMap database was used for small molecule drugs screening based on 20 real hub genes associated with JDM. Based on the criterion that the number of instances exceeds five and *P*-value less than 0.05, twelve small-molecule compounds were identified (Table 2). Among these compounds, acacetin, helveticoside, lanatoside C, deferoxamine, famprofazone, tanespimycin and LY-294002 may perturb the development of JDM, while betonicine,
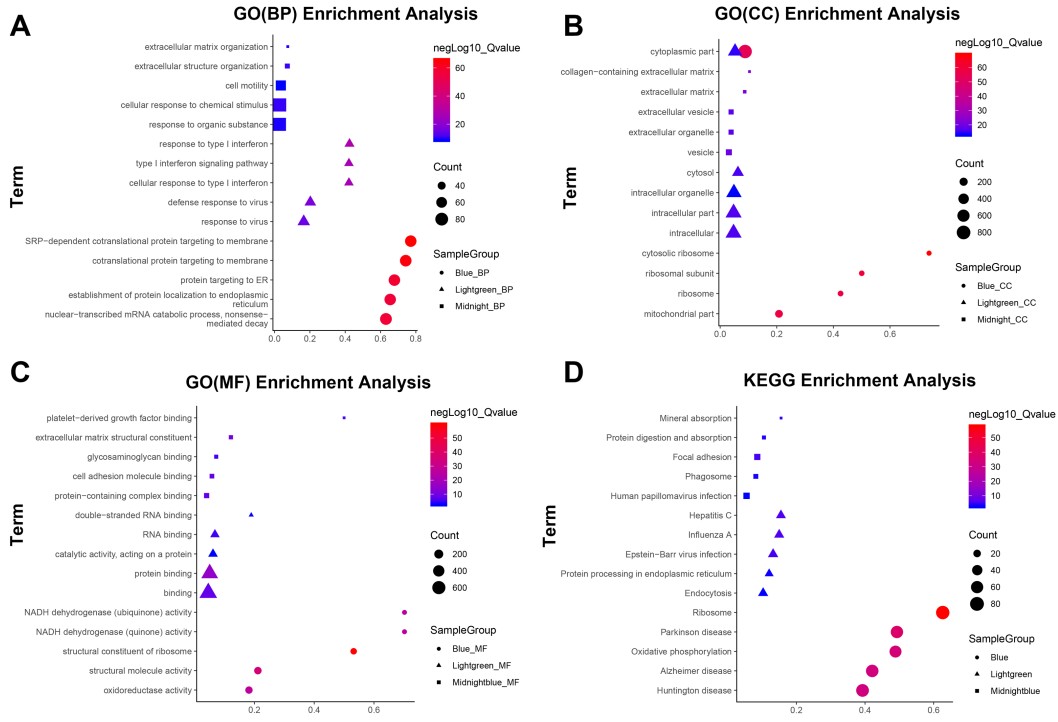

**Figure 3** **Top five terms of GO and KEGG pathway enrichment analysis in blue, light green and midnight blue modules.** The horizontal axis represents the gene ratio, and the vertical axis indicates the term of GO/KEGG signaling pathway, and the change of color from blue to red indicates the change of significance from low to high. (A) Biological function; (B) cellular component; (C) molecular function; (D) KEGG enrichment pathway.

felodipine, valproic acid, and sirolimus might provide potentially therapeutic goals for JDM.

## DISCUSSION

In this study, we used WGCNA to construct a co-expression network, detect key gene modules and identify hub genes in JDM for the first time. Our research provides some potential biomarkers or molecular targets for JDM through the Cmap database. We found that three modules highly correlated with JDM. The expression of 28 genes in these three modules showed significant changes in patients with JDM compared to control individuals in the training period, and 20 genes were validated as the real hub genes in the GSE11971 dataset, including the downregulation of NADH dehydrogenase, ATP synthase and cytochrome c oxidase and upregulation of IFN-stimulated genes. However, few of them were identified as biomarkers or crucial genes in JDM yet.

Functional enrichment analysis indicated that type I interferon signaling and various virus infection pathways were strengthened in JDM compared to HC, which is consistent with findings of previous studies (*Moneta, Marafon & Marasco, 2019*; *Piper et al., 2018*). IFIT5, IFI16 and MX2, interferon-stimulated genes, both nuclear transcriptional factors, were found to be upregulated in other autoimmune diseases but not in JDM (*Wang et*

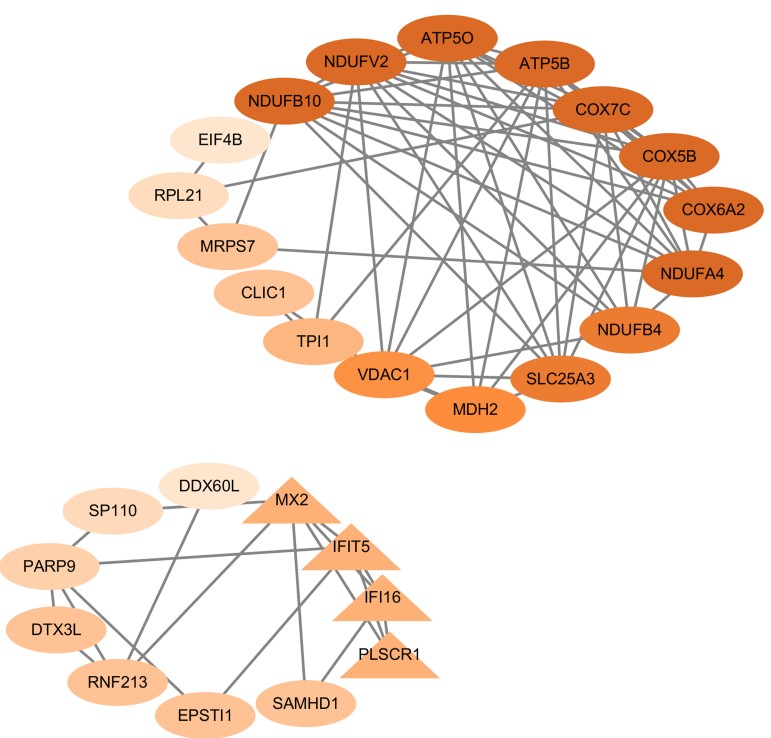

**Figure 4** **Protein–protein interaction network of 45 hub genes.** Different shapes represent various clusters constructed by molecular complex detection (MCODE). i.e., ellipses represent cluster 1 and triangles represent cluster 2. The color changes from dark brown to light brown indicate the MCODE score changes from high to low.

*al., 2019b*; *Zhang & Xu, 2019*). In the present study, we found that interferon-stimulated genes were significantly upregulated in JDM patients, as well as the so-called "interferon signature", demonstrating a possible mechanism that viral mimics or other stimuli may play a crucial role in the pathogenesis of JDM. Viral mimics are thought to participate in the pathogenesis of JDM (*Musumeci & Castrogiovanni, 2018*) and other autoimmune diseases (*Christen et al., 2004*; *Sellami et al., 2019*), consistent with the notion that JDM patients have higher rates of viral infections (*Tansley, McHugh & Wedderburn, 2013*; *Zheng et al., 2019*). This may suggest that the prevention of certain viral infections would decrease the incidence of autoimmunity by inhibiting self-antigenic mimics. NADH dehydrogenase (NDUFA4 and NDUFB4), ATP synthase (ATP5O and ATP5B) and cytochrome c oxidase (COX) family (COX5B, COX6A2 and COX7C) are crucial molecules involved in the oxidative phosphorylation in mitochondrial metabolism, and the decreased levels of these molecules suggested a crucial role of impaired mitochondrial phosphorylation and lower oxidative capacity in the pathogenesis of JDM, accounting for the extremity weakness in JDM patients.

Hypoxia caused by suppressed oxidative phosphorylation induces changes in reactive oxygen species (ROS) generation, whereby severe hypoxia in skeletal muscle results in elevated $H_2O_2$ generation. ROS accumulation produced by mitochondrial dysfunctions,

**Table 1** Hub genes identified by co-expression and MCODE methods.

| Gene | GS.JDM | MM | MCODE_Score | Regulation | Module |
|------|--------|-----|-------------|------------|--------|
| ATP5O | −0.860 | 0.905 | 7.0 | down | blue |
| ATP5B | −0.903 | 0.892 | 7.0 | down | blue |
| NDUFV2 | −0.792 | 0.912 | 7.0 | no sig | blue |
| NDUFB10 | −0.844 | 0.946 | 7.0 | NA | blue |
| COX7C | −0.784 | 0.919 | 7.0 | down | blue |
| COX5B | −0.830 | 0.937 | 7.0 | down | blue |
| NDUFB4 | −0.805 | 0.922 | 6.0 | down | blue |
| COX6A2 | −0.841 | 0.922 | 7.0 | down | blue |
| NDUFA4 | −0.833 | 0.833 | 7.0 | down | blue |
| VDAC1 | −0.880 | 0.896 | 4.8 | down | blue |
| SLC25A3 | −0.859 | 0.903 | 6.0 | down | blue |
| MDH2 | −0.851 | 0.921 | 5.0 | down | blue |
| TPI1 | −0.899 | −0.899 | 2.7 | down | blue |
| RPL21 | −0.827 | 0.915 | 0.5 | down | blue |
| MRPS7 | −0.842 | 0.916 | 2.0 | no sig | blue |
| PARP9 | 0.844 | 0.950 | 1.2 | NA | lightgreen |
| DTX3L | 0.844 | 0.961 | 2.0 | NA | lightgreen |
| MX2 | 0.930 | 0.913 | 3.0 | up | lightgreen |
| SAMHD1 | 0.846 | 0.954 | 2.0 | up | lightgreen |
| RNF213 | 0.841 | 0.961 | 2.0 | NA | lightgreen |
| EIF4B | −0.885 | −0.919 | 0.0 | down | lightgreen |
| IFIT5 | 0.934 | 0.921 | 3.0 | up | lightgreen |
| SP110 | 0.864 | 0.963 | 0.7 | up | lightgreen |
| IFI16 | 0.881 | 0.913 | 3.0 | up | lightgreen |
| PLSCR1 | 0.879 | 0.922 | 3.0 | up | lightgreen |
| EPSTI1 | 0.797 | 0.925 | 2.0 | NA | lightgreen |
| DDX60L | 0.846592 | 0.923402 | 0.0 | NA | lightgreen |
| CLIC1 | 0.768 | 0.933 | 2.0 | up | midnightgreen |

in turn, drives type I interferon responses and muscle inflammation, and may thereby self-sustain the disease process (*Wang et al., 2019a*). Similar to other autoimmune diseases, high-dose glucocorticoids, used alone or in combination with immunosuppressive agents are routine treatment for JDM patients wheras some refractory patients may develop functional limitations. It has been suggested that refractory JDM patients, have lower maximal oxygen uptake (*Drinkard et al., 2003*; *Hicks et al., 2002*) than do healthy children and with children with juvenile dermatomyositis in remission (*Takken et al., 2008*), suggesting that mitochrondrial dysfuction may contribute to the severity of JDM. Current concepts on the therapy of muscle weakness in JDM focus on induction of partial recovery and exposure to serious adverse events (including muscular toxicity). Our data suggest a novel therapeutic perspective for JDM by protecting mitochondria from dysfunction.

Bioinformatics combined human and material resources to develop more efficient tools with lower error rates (*Irizarry et al., 2003*). WGCNA is an efficient approach to construct

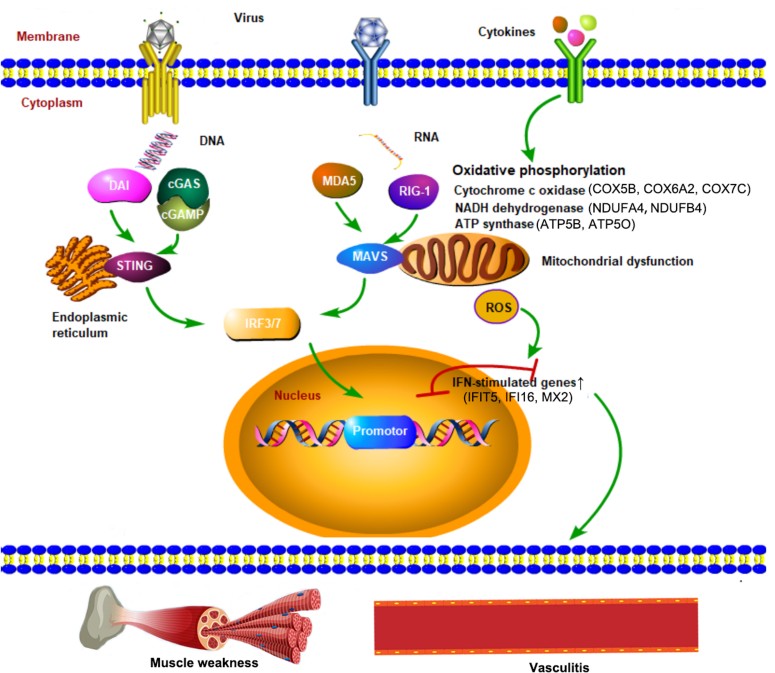

**Figure 5** **Cross-talk pathways involved in JDM interfered by hub genes and reference.** Enviromental factors (such as the invasion of virus) stimulate receptors to pathogen-associated molecular patterns (PAMPs) or cytokines on skeletal muscle cells and vascular endothelial cells, followed by the activation of DNA-cGAS-STING axis or RNA-MDA5/RIG-I-MAVS axis, thereby inducing the transcription of interferon-stimulated genes and other genes by phosphorylating IRF3/7, ultimately leading to inflammatory infiltration in blood vessels as well as skeletal muscle weakness. The dysfunction of metabolic signallings through cytokine receptors (for example, the type I interferon receptor (IFNAR) and/or IL-1 receptor (IL-1R)) activated oxidative phosphorylation in mitochondria, leading to the accumulation of reactive oxygen species (ROS), potentially resulting in energy-generating deficits in skeletal muscles. The hub genes involved in the pathogenesis of JDM are as follows: IFN-stimulated genes: IFIT5, IFI16 and MX2; Cytochrome c oxidase: COX5B, COX6A2 and COX7C; NADH dehydrogenase: NDUFA4 and NDUFB4 ; ATP synthase: ATP5B and ATP5O.

co-expressed modules and hub genes in several diseases. Previous studies using microarray expression profiles from adult-onset DM patients showed that IFN-stimulated genes were upregulated (i.e., MX2, STAT1 and OAS3), suggesting that the IFN signature overlapped the pathogenesis both in adult and juvenile DM. Nevertheless, mechanisms linked to hypoxia are less prevalent in adult-onset DM, suggesting mitochondrial dysfunctions contribute more to juvenile-onset DM rather than adult-onset DM.

We used the CMap database to predict several kinds of small-molecule compounds with promising capacity as therapeutic goals or inhibitors on treatment for JDM. No evidence has demonstrated the direct association between these compounds and JDM, while they hinted indirect link to JDM, according to the literatue. Among these compounds, acacetin, helveticoside, lanatoside C, deferoxamine, famprofazone, tanespimycin and LY-294002 showed negative enrichment scores and thus may have the potential to perturb the development of JDM, while betonicine, felodipine, valproic acid, and

**Table 2  Small molecule drugs screening based on CMap database.**

| Cmap name and cell line | Mean score | Number | Enrichment | P-value | Specificity | Percent non-null |
|---|---|---|---|---|---|---|
| 0175029-0000 | −0.696 | 6 | −0.982 | 0 | 0 | 100 |
| acacetin | −0.465 | 6 | −0.785 | 0.00018 | 0 | 100 |
| helveticoside | −0.494 | 6 | −0.754 | 0.00044 | 0.013 | 83 |
| lanatoside C | −0.424 | 6 | −0.7 | 0.00167 | 0.0638 | 66 |
| betonicine | 0.239 | 6 | 0.676 | 0.00312 | 0.0065 | 66 |
| deferoxamine | −0.345 | 8 | −0.556 | 0.00726 | 0.0184 | 62 |
| famprofazone | −0.203 | 6 | −0.557 | 0.02862 | 0.1035 | 50 |
| felodipine | 0.213 | 7 | 0.511 | 0.03088 | 0.1089 | 71 |
| trichostatin A - HL60 | 0.283 | 34 | 0.623 | 0 | 0.0798 | 52 |
| trichostatin A - MCF7 | 0.172 | 92 | 0.399 | 0 | 0.673 | 56 |
| LY-294002 - PC3 | −0.243 | 12 | −0.494 | 0.00325 | 0.2249 | 58 |
| valproic acid - HL60 | 0.254 | 14 | 0.417 | 0.01034 | 0.1812 | 50 |
| sirolimus - HL60 | 0.305 | 10 | 0.534 | 0.00359 | 0.0347 | 70 |
| vorinostat - MCF7 | 0.22 | 7 | 0.5 | 0.0378 | 0.7655 | 71 |

sirolimus showed positive enrichment scores and might provide potentially therapeutic goals for JDM. Acacetin, an inhibitor of lipopolysaccharide-induced inflammation, can promote the expansion of Treg cells and supress the differentiation of Th17 cells in a dose-dependent manner in collagen-induced arthritis (*Liu et al., 2018*). Helveticoside can regulate metabolism and signaling processes as a biologically active component, but little is known in inflammatory reactions (*Kim, Lee & Kim, 2015*). The iron chelator deferoxamine was shown to reduce mitochondrial oxidative stress in a transient cerebral ischemia model as well as the release of pro-inflammatory molecules including matrix metalloproteinase-9 and hypoxia inducible factor-1 (*Im et al., 2012*). LY294002, a kind of PI3K inhibitor, has potential against experimental autoimmune myocarditis (*Liu et al., 2016*). The heat-shock protein 90 inhibitor tanespimycin has been shown to inhibit cutaneous inflammation in experimental epidermolysis bullosa acquisita (*Tukaj et al., 2017*) and other experimental autoimmune models (*Dello Russo et al., 2006*). Felodipine, commonly used to treat hypertension and angin, has been evidenced to inhibit oxidative stress and inflammation in endothelial cells, which is consistent with our results (*Qi et al., 2017*). Valproic acid is a histone deacetylase inhibitor (HDACI), can suppress the inflammatory responses mediated by cytokines, oxidative stress molecules (ROS, NO), activating receptors (NK, T $\gamma$ $\delta$, and cytotoxic lymphocytes), perforin, granzyme, costimulatory molecules, and autoantibodies (*Soria-Castro et al., 2019*). Sirolimus can restore immune balance in rheumatoid arthritis patients by expanding the pool of circulating Treg cells (*Niu et al., 2019*). Our results based on the CMap database might provide hints as to future therapy for JDM; nevertheless, studies in vitro and in vivo are necessary.

This study has some limitations. First, this is retrospective, with all data in this study being retrieved from a public database. A multicenter, prospective study is needed to evaluate the significance of these hub genes in terms of long-term outcomes and possible

applications of molecular drugs for therapy. Second, experiments in vivo and in vitro are necessary to interpret potential mechanisms of real hub genes and small-molecule compounds for future clinical translation. Third, clinical traits cannot correlate with gene modules when performing WGCNA because of lack of clinical trait data in these GEO datasets.

## CONCLUSIONS

Based on weighted gene co-expression analysis, three key modules and 20 real key genes associated with the pathological state of JDM were identified, suggesting pivotal roles of mitochondrial dysfunction and the interferon signature in JDM. This analysis provides several candidate small-molecule compounds for use as targeted therapy of JDM.

## ACKNOWLEDGEMENTS

We would like to acknowledge the GEO, g:Profiler, STRING and CMap databases for free use.

### Funding

This work was supported by Chinese National Key Technology R&D Program of Ministry of Science and Technology (2017YFC0907604), National Science and Technology Major Project of the Ministry of Science and Technology of China (2019ZX09734001-002-004), Medical and health science and technology innovation project of Chinese Academy of Medical Sciences (2019-I2M-2-008), National Natural Science Foundation of China (81601430, 81471615). The funders had no role in study design, data collection and analysis, decision to publish, or preparation of the manuscript.

### Grant Disclosures

The following grant information was disclosed by the authors:
Chinese National Key Technology R&D Program of Ministry of Science and Technology: 2017YFC0907604.
National Science and Technology Major Project of the Ministry of Science and Technology of China: 2019ZX09734001-002-004.
Chinese Academy of Medical Sciences: 2019-I2M-2-008.
National Natural Science Foundation of China: 81601430, 81471615.

### Competing Interests

The authors declare there are no competing interests.

### Author Contributions

- Danli Zhong conceived and designed the experiments, performed the experiments, analyzed the data, prepared figures and/or tables, authored or reviewed drafts of the paper, and approved the final draft.

- Chanyuan Wu performed the experiments, authored or reviewed drafts of the paper, and approved the final draft.
- Jingjing Bai analyzed the data, prepared figures and/or tables, and approved the final draft.
- Dong Xu and Xiaofeng Zeng analyzed the data, authored or reviewed drafts of the paper, and approved the final draft.
- Qian Wang conceived and designed the experiments, authored or reviewed drafts of the paper, and approved the final draft.

## Patent Disclosures

The following patent dependencies were disclosed by the authors:
    Data is available at GEO: GSE3307 and GSE11971.

## Data Availability

    The data is available at NCBI GEO: GSE3307 and GSE11971.

## Supplemental Information

Supplemental information for this article can be found online at http://dx.doi.org/10.7717/peerj.8611#supplemental-information.

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
