# Peer review of "Co-expression network analysis reveals the pivotal role of mitochondrial dysfunction and interferon signature in juvenile dermatomyositis"

_PeerJ, doi:10.7717/peerj.8611_

## Round 0.1 · original submission · Minor Revisions

I agree with the reviewers that the language issues blur the outcome. All structural, grammar etc. matters should be carefully addressed in order to allow the readers to enjoy reading. In this context I would also point to the use of the word 'weighted' instead of 'weighed'. This also should be corrected.

Apart from other suggestions by the reviewers (especially comparison with other dermatomyositis types), I would stress the need to add a figure that puts your findings into context. Maybe a drawing of a mitochondrium and functions of proteins of interest shown in metabolic pathways? Any method to show the big picture would be great.

Reviewer 1 ·

Basic reporting

The article is quite well written, has a good structure. English is good, despite a few minor typos. Below I list a few minor concerns:

Referring to module colours in the abstract can be confusing. Similarily across the manuscript - referring to module colour seems unnatural. Maybe it would be worth to handle modules in another way? Besides information on module colour, no other data on the particular module's structure is available directly from the manuscript.

Line 117: It is hard to note those 12 modules in the picture - are there really 12 colours but mixed together?

Urgent: The figures lack readability. Fonts are too small and impossible to read. Figure fonts are also inconsistent across the manuscript.

Line 133: Misspelling: Go -> GO

Line 134: Biological process instead of BP would be more informative

Fig 4 should be moved to supplementary data in this form. Otherwise, the authors need to try to show it in a more feasible way suitable for the manuscript.

Line 86: Missing citation for KEGG

Fig 2A - The figure would benefit much from aligning tree's leaves with heatmap.

The paragraph starting at 132 - the long lists of GO terms are redundant with the figure. After tweak readability of the figure, the paragraph can be cleaned from redundant information.

What is the colouring scheme in Suppl fig 3?

Drug screening is missing from the Methods section. The reader would find it helpful to have a screening method underlying the cited database explained in a few sentences.

The paragraph starting at line 210: How do these drugs' mechanisms of action relate to the identified hub genes? Can authors at least try to rationalize why these drugs were chosen in the context of functional analysis of hub genes?

Experimental design

The reported study is well designed.

I'd like to ask the authors why only the GSE3307 dataset was used? Was that the only dataset (besides GSE11971, of course) with JDM data?

Also, did you compare obtained results to other dermatomyositis (not only juvenile) GEO data, which are available? Finding similarities/differences could be informative. Of course, this concern goes beyond the immediate scope of this study but maybe inspire the authors.

Validity of the findings

The findings seem to be sound in the light of performed analyses.

However, I miss the link between drugs' list and identified hub genes. Which of those genes are targeted by the drugs? How does the targeting affect them? Do we target up-regulated gene products? What are the mechanisms of proposed drugs' actions? I'd much appreciate some discussion on this.

·

Basic reporting

The manuscript by Zhong et al. is an example of in-depth in silco study using gene expression data available in public repositories, in this case of patients suffering from juvenile dermatomyositis. Such studies are well within the scope of PeerJ, and the reviewed manuscript has no fundamental flaws in methodology or interpretation of results. I do, however, have some significant concerns about the structure and language of the manuscript, that need to be addressed by an extensive rewriting before acceptance.
In general, the entire manuscript should be edited by a native speaker of English or a professional editing service, as it contains many examples of awkward phrasing and terminology, and incorrect grammar. I will list the most obvious language problems below, but they are too numerous to mention here.
Some sections of the manuscript should also be restructured and/or expanded, as explained below and in the subsequent sections of the review. Numbers refer to lines in the manuscript file

A. Language problems.
2. I have doubts about the word "signature" in the title. Did the authors mean "signalling"?
25. One does not screen a drug for genes, it's the gene products that are screened for potential small molecule inhibitors. Also, this sentence implies that the authors performed actual drug screening, while what they in fact did was search the database of known inhibitors using the genes identified in their study. There's nothing wrong with this, but it should be made clear in the text from the beginning.
61. "should contain" - did it, or did it not?
61-62. "specimens at least three biological replicates" - incorrect grammar/syntax
62. "package of limma" shold be "the limma package"
63. "The specific platform" - what platform? This isn't explained anywhere.
84. One does not use a database to upload genes. What the authors did was to search the database using the genes they found.
116. "equal" not "equals"
124-125. Awkward repetition of "independent"
133. "GO" not "Go"
162-163 and 187-189. The word "including" should not be used when all the genes are listed in the sentence.
169. "to for"?
172-173. Drugs don't "show negative correlation" with a disease. Meaning unclear.

B. Structure and completeness
16-18. This is not background. Background section of the abstract should summarize the introduction, i.e. explain the problem (JDM).
23. Don't put detailed methods (like the information that MCODE is a Cytoscape plugin) in the Abstract, just name the main methods used.
25-26. The colors assigned to the modules are arbitrary, here they are out of context. Use something like "Three key modules were identified".
29. See comment for lines 172-173 above.
58-66. What is described here is the critical stage in a gene expression analysis - identification of differentially expressed genes (DEGs). This should be divided into two sections: "Datasets" and "Normalization and DEG analysis". The "Dataset" section should contain the GEO IDs of the datasets with the number of JDM and healthy control samples in each (no need to list each sample in a dataset, this is correctly presented as supplemental data). Information about the platform used to obtain the data should also be included. Providing this information as a concise table would be optimal for legibility. Also, the normalization of data and DEG identification should be described more clearly. Was between array or within array normalization performed (or, preferably, both?), and was limma used for all the normalization steps (sva is often used for normalization prior to DEG analysis in limma)?
68-69 and 113. The list of top DEGs should be provided as a supplemental file.
106-111. This information should be in the first section of "Materials and methods", preferably in a table form (see comments for 58-66 above), and only briefly referenced here.
123-167. It would be good to provide lists of GO terms and KEGG pathways for identified modules (JDM and control), as well as a list of hub genes. In general - all the lists identified in the study should be made available as supplemental information.
168-173. See comment in section A - explain what is the correlation for the three compounds and how they are different from the remaining 8 in Table 2 with respect to their action on the identified hub gene products.
176-182. Two unrelated statements (a description of JDM and a description of WGCNA algorithm) should not form a single paragraph. Also, this information belongs in the Introduction, rather than Discussion.
183. See comment for lines 25-26. The discussion should be readable without going back to figures or results, and the colors are meaningless out of context.
187-190. This is an exacct copy of the information already presented in lines 161-165 and in Table 1, a reference to Table 1 would be sufficient.
232. A description of author contributions is customarily provided here.

Experimental design

This research is within the aims and scope of PeerJ. Its strengths include a rich dataset and in-depth statistical analysis. The use of DEG co-expression networks combined with knowledge contained in GO and KEGG databases is a routine method that often yields significant insights. This study is no exception, and the obtained results provide insights that were not apparent when the original datasets were published. It is therefore an excellent example of the possibilities afforded by using modern bioinformatic and statistical tools with the wealth of data contained in public repositories. The biomedical problem selected for the study is well defined and relevant, and the study is performed to a sufficiently high technical standard.
The methods section should be expanded. In addition to comments provided in the previous section of the review, I also believe that it would be beneficial to provide snippets of R code used to perform the analyses in supplemental files.

Validity of the findings

I believe the data used in this study are robust and adhere to the standards used in the fiels. In the comments contained in the first section of the review I stated what data should be additionally included as supplemental files.
The discussion is brief and to the point, I would even encourage some more speculation and exploration. For example, the authors link the mitochondrial respiratory chain genes identified in the study to ROS production. Do they believe that mitochondrial dysfunction is among the causes of JDM, or is it rather a result of the JDM pathology? Is the ROS hypothesis supported by the data, are there any genes related to oxidative stress in the lists identified in the study?

Additional comments

The negative comments are generally related not to the methods ad results of the study, but rather to the way it is presented and described.
A recent study published in PeerJ from the same institution (Chen S, Yang D, Lei C, Li Y, Sun X, Chen M, Wu X, Zheng Y. 2019. Identification of crucial genes in abdominal aortic aneurysm by WGCNA. PeerJ 7:e7873 https://doi.org/10.7717/peerj.7873) could serve as a model to show how the majority of my concerns could be addressed, and what methods and data should be included in supplemental fles. If there is a significant overlap in the methods used in these two studies, a reference to the published methods could replace some information in your methods section (something like "array data normalization was performed as in Chen et al. 2019" if applicable),

---

## Round 0.2 · Minor Revisions

Revisions suggested are minuscule therefore you may regard it as accepted. Still, take into account Reviewer's 1 suggestions as well as modify Figure 5 please. Since I'm the cause of this great overview I would suggest a clearer explanation around the 'IFN-stimulated genes' area of the drawing. The arrows are not self-explanatory in this section. Also, add the names of hub genes in the appropriate sections in order for it to be fully synchronized with the description.

I find your improvements significant and it seems that we are close to the end of this procedure.

Good luck!

Reviewer 1 ·

Basic reporting

I'd like to thank the authors for significantly improving the manuscript, especially for adding the paragraphs on biological interpretation. I found however few very minor issues, which could be addressed before final accepting the manuscript. (I include also typos and small editorial issues I found in order to help a little bit).

Lines 32-34 (abstract) and one more in the manuscrpit: what is the difference between the potential to perturb and providing therapeutic goals? These two categories contain different substances but I cannot get the difference.

Line 51: JDM children - shouldn't be JDM in children? Besides, JDM means juvenile so why children?

Line 57: Missing whitespace before Adaptative

Line 61: method that mines (the s form)

Line 139: Please define DEG once more - not everyone (including me) reads methods section.

Lines 139-140: based on [...] 0.05, - these detailes should be placed in the methods section rather than results...

Line 143: We input - OK, but input where?

Line 148 and following: could that be possible to put the numbers into the figure? The paragraph looks awkward in this form...

Line 226: missing 'e' in suggested

Line 250: used the databse TO predict

Line 253: according to literature - please provide some refs then.

Table 1: why keeping so many decimals for VDAC1 and SP110 while not for e.g. TPI1? I'd suggest rounding to the first decimal. Also round the numbers for DDX60L to the 3rd?

Table 2: I'd enjoyed this table much more knowing what those columns mean. It was explained in the methods, but the reader would benefint greatly if the caption carried some brief information, e.g. "Mean in the range xxx-yyy mean that the target is ...", N and P also look abstract.

Figure 4: Please consider dragging and dropping the objects, especially in the triangle-including part. Circle-like representation is quite confusing and is isn't imposed by the data type. After a little bit of focus the reader will find the denser clusters himself, but why not helping him a little? Let the clusters form clusters - Cytoscape allows for moving objects.

Figure 2: While studying this figure I thought in that in the tree maybe instead of colour names the real colours could be used? Or you could simply skip tree leaf names and moved the color legend in the matrix below from the below to the above to serve as legend for two figures at once?

Figure 1: leaf names are unreadable in the tree.

REMARK ON COLOURS: First of all - you know which clusters are chosen and presented individually on further figures. Why not choosing more discerning colour for those instead of blue, lightgreen and midnightblue - all of them similar in fact? Maybe orange, teal (of some pastel flavour) and light green - they separate visually? You could easily use those colours across the figures and the data would be much more readable. The rest of the clusters also looks quite dull, but they are rather for informative purposes as they are not chosen further.

Overall - good job!

Experimental design

No comments for now.

Validity of the findings

No comments for now.

Additional comments

I'm giving 'minor revision' to give the authors a chance to tweak the manuscript a little. Yet I'm satisfied with the revision.

·

Basic reporting

The revised manuscript is a significant improvement of the original submission. It is now structured in a way that is acceptable for publication.

Experimental design

The revision improved the description of methods and results, and is now sufficient to warrant publication.

Validity of the findings

The revision does not change my initial favourable opinion on the validity of findings.

Additional comments

I believe the paper is significantly improved in this revision and can be accepted for publication in PeerJ.

---

## Round 0.3 · accepted · Accept

Great improvements, great work, and good luck! :)